Antifibrotic effect of xanthohumol in combination with praziquantel is associated with altered redox status and reduced iron accumulation during liver fluke-associated cholangiocarcinogenesis

Jamnongkan Wassana 1 2
Thanee Malinee 1 2
Yongvanit Puangrat 1 2
Loilome Watcharin 1 2
Thanan Raynoo 1 2
Kimawaha Phongsaran 2 3
Boonmars Tidarat 2 4
Silakit Runglawan 1 2
Namwat Nisana 1 2
Techasen Anchalee anchte@kku.ac.th anchaleetechasen@gmail.com 2 3
1 Department of Biochemistry, Faculty of Medicine, Khon Kaen University , Khon Kaen , Thailand
2 Cholangiocarcinoma Research Institute, Khon Kaen University , Khon Kaen , Thailand
3 Faculty of Associated Medical Sciences, Khon Kaen University , Khon Kaen , Thailand
4 Department of Parasitology, Faculty of Medicine, Khon Kaen University , Khon Kaen , Thailand
Yamamoto Tadashi
Electronic publication date: 2018 Jan 22
Publication date: 2018
Volume: 6
Electronic Location ID: e4281
Received 2017 Aug 28; Accepted 2017 Dec 29
Copyright: ©2018 Jamnongkan et al.
Copyright year: 2018
Copyright holder: Jamnongkan et al.
License: This is an open access article distributed under the terms of the Creative Commons Attribution License, which permits unrestricted use, distribution, reproduction and adaptation in any medium and for any purpose provided that it is properly attributed. For attribution, the original author(s), title, publication source (PeerJ) and either DOI or URL of the article must be cited.
License URL: https://creativecommons.org/licenses/by/4.0/

Keywords: Cholangiocarcinoma, Oxidative stress, DNA damage, Iron, Chemoprevention, Fibrosis

Funding: Thailand Research Fund Thailand Research Fund and Khon Kaen University Faculty of Medicine IN58246 This study was supported by the Thailand Research Fund through the Royal Golden Jubilee Ph.D. Program (to Wassana Jamnongkan and Puangrat Yongvanit), the Thailand Research Fund and Khon Kaen University to Anchalee Techasen: TRG5880023. The Higher Education Research Promotion and National Research University Project of Thailand, Office of the Higher Education Commission, through the Center of Excellence in Specific Health Problems in Greater Mekong Sub-region cluster (SHeP-GMS), Khon Kaen University to Anchalee Techasen and a grant from the Faculty of Medicine (Grant No. IN58246), Khon Kaen University, Thailand. The funders had no role in study design, data collection and analysis, decision to publish, or preparation of the manuscript.

==============================
Cholangiocarcinoma (CCA) caused by infection of the liver fluke Opisthorchis viverrini, (Ov) is the major public health problem in northeast Thailand. Following Ov infection the subsequent molecular changes can be associated by reactive oxygen species (ROS) induced chronic inflammation, advanced periductal fibrosis, and cholangiocarcinogenesis. Notably, resistance to an activation of cell death in prolonged oxidative stress conditions can occur but some damaged/mutated cells could survive and enable clonal expansion. Our study used a natural product, xanthohumol (XN), which is an anti-oxidant and anti-inflammatory compound, to examine whether it could prevent Ov-associated CCA carcinogenesis. We measured the effect of XN with or without praziquantel (PZ), an anti-helminthic treatment, on DNA damage, redox status change including iron accumulation and periductal fibrosis during CCA genesis induced by administration of Ov and N-dinitrosomethylamine (NDMA) in hamsters. Animals were randomly divided into four groups: group I, Ov infection and NDMA administration (ON); group II, Ov infection and NDMA administration and PZ treatment (ONP); the latter 2 groups were similar to group I and II, but group III received additional XN (XON) and group IV received XN plus PZ (XONP). The results showed that high 8-oxodG (a marker of DNA damage) was observed throughout cholangiocarcinogenesis. Moreover, increased expression of CD44v8-10 (a cell surface in regulation of the ROS defense system), whereas decreased expression of phospho-p38MAPK (a major ROS target), was found during the progression of the bile duct cell transformation. In addition, high accumulation of iron and expression of transferrin receptor-1 (TfR-1) in both malignant bile ducts and inflammatory cells were detected. Furthermore, fibrosis also increased with the highest level being on day 180. On the other hand, the groups of XN with or without PZ supplementations showed an effective reduction in all the markers examined, including fibrosis when compared with the ON group. In particular, the XONP group, in which a significant reduction DNA damage occurred, was also found to have iron accumulation and fibrosis compared to the other groups. Our results show that XN administered in combination with PZ could efficiently prevent CCA development and hence provide potential chemopreventive benefits in Ov-induced cholangiocarcinogenesis.

Introduction

Chronic inflammation induced by liver fluke (Opisthorchis viverrini, Ov) infection is the major risk factor for cholangiocarcinoma (CCA) in northeast Thailand. Re-infection of people in endemic areas occurs more likely after praziquantel treatment and frequently leads to periductal fibrosis that governs the pathogenic effects of inflammation and immunity in liver fluke-associated cholangiocarcinogenesis has been recently reviewed (Yongvanit, Pinlaor & Bartsch, 2012). Cellular damage caused by reactive oxygen species (ROS) and reactive nitrogen species (RNS) during chronic infection appears to be a key event which is further enhanced by an imbalanced oxidant/anti-oxidant system during reinfections (Yongvanit et al., 2012). Increase in periductal fibrosis has been shown to be associated with CCA genesis (Prakobwong et al., 2009). After Ov infection, time-dependent periductal fibrosis has been detected with subsequent CCA development (Chamadol et al., 2014; Mairiang et al., 1992).

Recently, studies have shown that the expression of CD44 variants form (CD44v) could stabilize xCT (cysteine-glutamate transporter) which promotes cystine uptake and then its conversion to cysteine contributing to glutathione synthesis for ROS defense (Ishimoto et al., 2011; Kim et al., 2002). Moreover, Thanee et al. (2016) showed that an accumulation of CD44v8-10 causes suppression of phospho-p38MAPK, a major ROS target expression in transforming bile duct cells which is linked to a poor prognosis in CCA patients. In addition, the regulation of redox status in CCA cell lines depends on the expression of CD44v8-10. Therefore, CD44v8-10 plays an important role in redox status regulation via stabilizing xCT in CCA development. During CCA genesis, chronic inflammation-led oxidative stress may induce gene related redox status regulation including CD44v to protect bile duct epithelial cells from ROS, hence facilitating CCA genesis.

Additionally, iron is an essential nutrient and the most abundant transition metal in human body, where under physiological conditions it exists in its stable redox states, ferrous ion (Fe2+) and ferric ion (Fe3+) (Domaille, Que & Chang, 2008). The essential roles of iron in various biological events, such as oxygen delivery (Wilson & Reeder, 2008), electron transport (Rouault & Tong, 2005), and enzymatic reactions (Aisen, Wessling-Resnick & Leibold, 1999; Costas et al., 2004), depend on its redox activity. On the other hand, iron overload causes severe cell damage and organ dysfunction through the abnormal production of ROS (Halliwell & Gutteridge, 1992; Xu et al., 2012). Thus, iron plays critically important roles in both healthy and diseased states of the living organisms. The role of iron in carcinogenesis has been shown to be associated with oxidative DNA damage (Wiseman & Halliwell, 1996). We have previously reported that transferrin receptor-1 (TfR-1), a cell surface receptor is a candidate molecule for involvement in the increasing of cellular iron uptake in CCA (Jamnongkan et al., 2017). In addition, an increase in TfR-1 has also shown to be responsible for transferrin-mediated iron uptake occurs in breast (Pinnix et al., 2010) and pancreatic cancers (Ryschich et al., 2004).

Generally, the gold standard treatment for CCA is surgical resection. However, complete resection is often impossible, and eventually when surgery is achieved, is typically followed by metastasis and/or local recurrence (Khuntikeo et al., 2014; Titapun et al., 2015). The other treatment options such as chemotherapy and radiotherapy have been ineffective for the patients with inoperable tumors. Recently, extensive research has attempted to identify the effective nontoxic nutrients, phytochemical and synthetic pharmacological agents which are believed to have the ability to delay the onset of the carcinogenesis process. Curcumin, an anti-inflammatory agent, has been reported in many studies as a potential chemopreventive agent in CCA but the results are quite controversial. So far, it has been shown to inhibit an inflammatory reaction in Ov infected hamsters, but it cannot inhibit Ov-induced CCA in the hamster animal model (Boonjaraspinyo et al., 2009; Pinlaor et al., 2010; Prakobwong et al., 2009). Therefore, other chemopreventive agents need to be identified and explored for their effectiveness in CCA prevention and/or treatment.

Xanthohumol (XN) is a principal flavonoid of hop plant (Humulus lupulus L.) which has been isolated from hop cones, that are largely used in the brewing industry as a preservative and flavoring agent to add bitterness and aroma to beer (De Keukeleire et al., 1999; Taylor et al., 2003). It has been identified as having anti-inflammation, anti-oxidant and cancer chemopreventive activities (Gerhauser, 2005). Moreover, XN has been shown to be associated with induction of apoptosis via increased p21 and p53 expression and decreased surviving levels in a leukemia cell line (Monteghirfo et al., 2008). Additionally, it reduced ROS formation (Hartkorn et al., 2009) and protection against DNA damage (Pinto et al., 2012). Dokduang et al. (2016) reported that XN effectively suppressed the growth of tumor and induced apoptosis in CCA cells and tumor inoculated mice by which inhibiting STAT3 activation due to suppression of the Akt-NFκB signaling pathway. These data suggested that XN has a potential to become a useful new approach for the chemoprevention and/or treatment of CCA. Additionally, praziquantel (PZ) is well known as a potential drug for liver fluke treatment, which has also been used for removal of Ov. The cure rate of PZ treatment in opisthorchiasis may be as high 100% effective in both humans (Bunnag & Harinasuta, 1981; Bunnag et al., 1984) and hamsters (Duenngai et al., 2013). Decreasing levels of liver fibrosis and egg granulomas have been revealed in the liver of patients with blood and liver fluke infections after PZ treatment (Pinlaor et al., 2010).

Therefore, this study investigated the effect of XN with or without PZ, on DNA damage, redox status change and periductal fibrosis during CCA genesis in a hamster animal model.

Materials and Methods

Parasite preparation and animal infection

Ov metacercariae were extracted from naturally infected cyprinid fish, which were purchased from the market in Khon Kaen Province, northeast Thailand. The fish were minced and digested with pepsin-HCL, then incubated in shaking water bath at 37 °C for an hour. The digested fish were filtered through the sieves (1,000, 425 and 106 μM respectively) and the filtrate was sedimented with 0.85% saline in a sedimentation jar. Finally, the metacercariae were isolated and identified under a stereomicroscope. Fifty viable active cysts were fed to each of five male Syrian golden hamsters by intragastric intubation.

Xanthohumol preparation

XN was kindly provided by Hopsteiner, Mainberge, Germany. XN-supplemented water was prepared daily as 1 μl of stock 20 mM in 250 μl of distilled-water, yielding a final concentration of 20 μM or 171 mg/B.W./day in the assigned groups. Hamsters were pre-treated with XN more than 14 days before the experiment commenced and treatment was continued until animals were sacrificed at day 60, 90, 120 and 180 post-treatment.

N-nitrosodimethylamine preparation and administration

N-nitrosodimethylamine (NDMA) (Sigma-Aldrich, St. Louis MO, USA) was diluted in distilled water at 12.5 ppm, and administrated daily to the assigned hamster groups starting on day 30 until day 60 after Ov infection.

Praziquantel preparation

PZ (Sigma-Aldrich, St. Louis MO, USA) was diluted with 2% chemophor, a non-ionic solubilizer and emulsifier. A single dose of 400 mg/kg was administered orally to the assigned treatment groups after day 30 of Ov infection.

Animal groups

The Animal Ethics Committee of Khon Kaen University (AEKKU 23/2555) approved the study protocol. Six- to eight-week-old male Syrian golden hamsters were randomly divided into four groups as followings: group I, Ov infection and NDMA administration (ON); group II, Ov infection and NDMA administration and PZ treatment (ONP); groups III and IV were similar to group I and II, they received 20 μM XN (171 mg/B.W./day) designated as XON and XONP groups, respectively. Hamsters were treated with 50 Ov metacercaria by oral inoculation which was administered with 12.5 ppm of NDMA in water for 30 days and withdrawn thereafter. XN-supplemented water was prepared daily as 1 μl of stock 20 mM in 250 μl of distilled-water. The animals were sacrificed at 60, 90, 120 and 180 days after treatment and hamster liver tissues were collected for further analysis.

General observations

Body and liver weights of each hamster were evaluated at the scarification time. Data were expressed as mean ± SD of liver/body weight. In addition, hamsters in ON, XON, ONP, and XONP groups were used for survival analysis, and the number of surviving hamsters was counted monthly.

Gross observation

Four lobes of the liver (right, left, caudate, and quadrate) were examined for changes in color, appearance of the margins, presence or absence of nodules, and granularity of surfaces, both visually and from photographs taken with a digital camera.

Histological observation by Hematoxylin and Eosin (H&E) staining

Hamster liver tissues were sectioned at a thickness of 4 μm, then fixed in 10% buffered formaldehyde and embedded in paraffin. H&E staining was performed in the sections for histological observation. The sections of liver tissue were deparaffinized in xylene for 3 min for three times to remove the paraffin wax, and then rehydrated in a dilution series of ethanol. The sections were rinsed with tap water, stained with Harr’s hematoxylin for 10 min, and then washed with the running tap water for 2 min. Destaining in ethanol containing 1% hydrochloric acid was performed, then washed with running tap water, and stained in saturated lithium carbonate for 3–4 s. The sections mounted on slides were stained with eosin solution after washing with running tap water for 10–20 min. Dehydration of all sections was performed before mounting with permount. Transformation of bile duct cells including normal, hyperplasia, dysplasia and cholangiofibrosis, and cancerous lesions were examined under a light microscope (Axioscope A1; Carl Zeiss, Jena, Germany) at high magnification (×200).

Immunohistochemistry

Sections mounted on slides were deparaffinized in xylene and rehydrated in an ethanol concentration series. Three different methods were used to retrieve antigen from tissue sections; (1) for CD44 variant 8–10 staining; 0.5% trypsin-EDTA for 20 min at 37 °C incubator, (2) for phospho-p38MAPK and 8-oxodG staining, 0.01 M sodium citrate containing 0.05% Tween 20 (pH 6.0) was autoclaved for 10 min at 110 °C, and (3) for TfR-1 microwave treatment was performed in 10 mM citrate buffer pH 6.0 at high power for 10 min. After that, exposure of 3% H2O2 and 3% bovine serum albumin was performed before further incubation with primary and secondary antibodies, respectively. A Vectastain Elite Kit (Vector Laboratories, Burlingame, CA, USA) and 3,3′-diaminobenzidine was used for detecting the staining signal, and the sections were counterstained with Mayer’s hematoxylin. The stained sections were observed under a light microscope at high magnification (×400). The intensity of staining was evaluated semi-quantitatively by the Allred Scoring protocol (Allred et al., 1998).

Prussian blue staining

Prussian blue was used for localization of ferric iron in hamster liver tissues. Hamster liver sections were deparaffinized and rehydrated with graded alcohol. Slides were immediately transferred into a working staining solution (equal volumes of 10% potassium ferrocyanide and 10% hydrochloric acid) for 20 min at room temperature. Any ferric ions (Fe3+) in the tissue combines with the ferrocyanide and result in the formation of a bright blue pigment. Subsequently, slides were rinsed in distilled water, and then counterstained with nuclear-fast red. Finally, sections were dehydrated with stepwise increasing concentrations of ethanol, cleared with xylene and mounted with permount. The stained sections were examined under a light microscope.

Sirius red staining

Sirius red was used for collagen staining. Paraffin sections of each hamster group were dewaxed in xylene and rehydrated with graded alcohol. The sections were stained with haematoxylin and eosin for 8 min, washed in distilled water, incubated with 0.1% Sirius red solution dissolved in aqueous saturated picric acid for 1 h at room temperature, washed in acidified water, dehydrated in xylene, and mounted with mounting medium. The slides were analyzed by light microscope. Collagen, which is the component of fibrosis formation, stained red around the bile duct areas. Therefore, the thickness of periductal fibrosis is stained in red color. The evaluation of staining was adapted using the Batts-Ludwig and IASL for grading fibrosis (Goodman, 2007). Stage 0: no fibrosis; stage 1: mild fibrosis; stage 2: severe fibrosis.

Statistical analysis

Statistical analysis was performed using SPSS software. The difference in general observation results was performed by student’s t-test (nonparametric: Mann–Whitney test) in two-tailed test. P-values of <0.05 were considered as statistically significant.

Results

General observations

The liver per body weight of hamster groups either with or without XN and PZ were not significantly different between groups. Moreover, survival curve analysis showed no statistical differences in the time that hamsters survived between each group. However, it seems that the groups with XN treatment had longer survival times either in the presence or absence PZ treatment but this was not statistical significant.

Gross and histological appearances

Gross pathology of the livers revealed smoothness of liver surfaces, although slightly opaque around common bile ducts in all hamsters of both the ON and XON groups throughout the experimental period. White granules and small foci were found in the hepatic tissues at day 180 after treatment in both the ON and XON groups (Fig. 1). In addition, the obstruction of gallbladder was seen in the ON and XON groups but not in the ONP and XONP groups. However, no tumor mass was detected in the liver of all hamsters belonging to the above treatment groups.

Figure 1 Gross findings of hamster liver tissues.

Photographs are representative at days 60, 90, 120, and 180 in four groups including the control (ON), presence of xanthohumol (XON), presence of praziquantel (ONP) and presence of xanthohumol and praziquantel (XONP) groups. Results show the obstruction of the gallbladder in the ON and XON groups at 60, 90, 120, and 180 days but not in the ONP and XONP groups. Additionally, white granules were observed at day 180 in ON and XON groups. Arrowheads indicate white granules and small foci.

Histopathological findings revealed no changes in livers and bile ducts in the XN treatment alone, that was similar to untreated group, hence XN was not toxic to liver and bile duct cells. In the ON group, aggregation of inflammatory cells, dilated bile ducts as well as fibrosis was observed at all-time points. Additionally, massive hyperplasia and dysplasia of bile ducts was observed at day 60, 90, 120 and 180. One of the hamsters (1/5, 20%) developed CCA by day 180, whereas the others did not develop CCA. The most severe changes found in the rest of hamsters at day 180 were cholangio-fibrosis. Interestingly, there was no CCA development observed in the XON group. The most severe pathological changes observed at day 180 in this group was cholangio-fibrosis (4/5, 80%), while only bile duct proliferation and hyperplasia were seen in one of the hamster (1/5, 20%) as shown in Fig. 2. These results indicated that XN could repress the progression of CCA development. Importantly, there was no CCA development in either PZ treated CCA-induced hamster (ONP) or combination with XN treatment (XONP). The most severe pathological changes observed in these groups at day 180 was only bile duct hyperplasia.

Figure 2 Histological changes in Ov-induced CCA hamsters with the control (ON), presence of xanthohumol (XON), presence of praziquantel (ONP) and presence of xanthohumol and praziquantel (XONP) groups.

Circles represent: Ov, O. viverrini; BD, bile duct; IF, inflammation area; HD, hyperplasia bile duct; PD, proliferative bile duct.

Effect of XN with or without PZ supplements on DNA damage

8-oxodG was used as a marker of oxidative stress. A time profile of 8-oxodG formation in the liver of hamsters was evaluated as detected by immunohistochemical staining. The expression of 8-oxodG was observed mainly in the nucleus of bile ducts and inflammatory cells from day 60 and reached the highest on day 180 in the ON group (Fig. 3A). Interestingly, either PZ or XN treatment significantly reduced the accumulation of 8-oxodG starting from day 60 and 120, respectively, when compared with the ON group (P < 0.05). Moreover, our results showed that the XONP group had significantly and more effectively reduced DNA damage than other groups (P < 0.05) (Fig. 3B).

Figure 3 Immunostaining of 8-oxodG in CCA genesis hamsters.

Localization of 8-oxodG in the control (ON), presence of xanthohumol (XON), presence of praziquantel (ONP) and presence of xanthohumol and praziquantel (XONP) groups (A). Bar graph comparisons of frequency and intensity using Allred score in each group of hamster liver (B). *P < 0.05 compared to the ON group.

Effect of XN with or without PZ supplements on redox status

Our result showed that higher CD44v8-10 expression was seen along cholangiocarcinogenesis. The strongest signal of CD44v8-10 was found at day 180 in both dysplasia and malignant area of the bile duct epithelia in ON group (Fig. 4A). On the other hand, the highest signal of phospho-p38MAPK which is a downstream target of ROS signaling was seen in the nucleus of the hyperplastic bile duct cells at day 60 and then decreasing of phospho-p38MAPK level was seen along the progression of carcinogenesis in ON group (Fig. 4C). Furthermore, both protein levels of CD44v8-10 and phospho-p38MAPK were significantly reduced from day 60 in the ON group treated with either XN or PZ (XON and ONP) (P < 0.05) (Figs. 4B and 4D). However, no synergistic effect was observed in the XONP group.

Figure 4 Immunostaining of CD44v8-10; cancer stem-like cells (A) and phospho-p38MAPK; downstream of ROS signaling (C) in CCA genesis in hamsters.

Photographs are representative of the control (ON), presence of xanthohumol (XON), presence of praziquantel (ONP) and presence of xanthohumol and praziquantel (XONP) groups. Bar graph comparisons of frequency and intensity using the Allred score in each group of hamster liver (B and D). *P < 0.05 compared to ON group.

Effect of XN with or without PZ supplements on iron accumulation and TfR-1 expression

Prussian blue was used for localization of ferric iron in hamster liver tissues. In the ON group, iron accumulation was observed from 60 days until tumor developed at 180 days (Fig. 5). The highest accumulation of iron was found at day 180 in both malignant bile duct epithelia and inflammatory cells. In addition, immunohistrochemical staining for TfR-1 protein showed that TfR-1 was localized in bile duct membranes and inflammatory cells (Fig. 6A). The highest of its expression was observed in cancerous lesions at day 180 in the ON group. Interestingly, the reduction of both molecules was seen in XON and ONP groups at day 90 when compared with the ON group. Moreover, our results showed an effective reduction of iron accumulation and TfR-1 expression for the XONP group compared to all other groups. This was especially evident in the expression of TfR-1 in XONP group which was significantly reduced from day 90 (P < 0.05) (Fig. 6B).

Figure 5 Prussian blue staining of iron in CCA genesis hamsters.

Photographs are representative of the control (ON), presence of xanthohumol (XON), presence of praziquantel (ONP) and presence of xanthohumol and praziquantel (XONP) groups.

Figure 6 Immunohistochemistry for TfR-1 in CCA genesis hamsters.

Photographs are representative of the control (ON), presence of xanthohumol (XON), presence of praziquantel (ONP) and presence of xanthohumol and praziquantel (XONP) groups (A). Bar graph comparisons of frequency and intensity using Allred score in each group of hamster liver (B). *P < 0.05 compared to the ON group.

Figure 7 Sirius red staining for collagen in CCA genesis hamsters with the control (ON), presence of xanthohumol (XON), presence of praziquantel (ONP) and presence of xanthohumol and praziquantel (XONP) groups.

Thickness of red color represents fibrosis area.

The inhibitory effects of XN and PZ on fibrosis

The thickness of periductal fibrosis stained in red color is representative of severe or mild fibrosis. In the ON group, severe fibrosis was seen starting from day 60 until day 180. Additionally, treatment with either XN or PZ (XON and ONP groups) resulted in only mild fibrosis at days 180 and 120, respectively. This was especially so, in the combination treatment or XONP group, where only mild fibrosis was detected from day 60 as shown in Fig. 7.

Discussion

At present, the strategies for CCA prevention and treatment are still ineffective and identifying new targets for potential therapeutic prevention of carcinogenesis should be undertaken. A broad-spectrum of cancer chemopreventive agents which act by inhibiting multiple mechanisms for cancer development has been reported (Gerhauser et al., 2002). One of the interesting natural products for cancer prevention is XN. Many studies have indicated that XN can prevent hepatic inflammation and fibrosis during chronic liver diseases as well as the progression of liver cancer (Dorn, Heilmann & Hellerbrand, 2012). Additionally, XN could reduce ROS formation (Hartkorn et al., 2009), protection against DNA damage (Pinto et al., 2012) and regulation of apoptosis and suppression of hepatic stellate cell activation (Yang et al., 2013). These data suggested that the XN has the potential to become a useful new approach for the chemoprevention and/or treatment of CCA.

Therefore, we investigated the effect of XN supplement on DNA damage formation in a CCA-induced hamster animal model. The expression of 8-oxodG which is an oxidative stress marker was decreased in XN-treated groups. When used together with the PZ (XONP) group it was found to be more effective in reducing DNA damage leading to alteration of redox status and repression of CCA development. This result correlated with our gross and histological observations at day 180 of treatment, which demonstrated that 20% of the ON group developed CCA while in the remaing hamsters (80%) only cholangio-fibrosis was found. After XN was supplemented, 20% of hamsters had slow progression of CCA development because only bile duct proliferation and hyperplasia were seen in these groups. Importantly, there was no CCA development in combination with the PZ treatment (XONP). The most severe pathological changes observed in these groups were only bile duct hyperplasia.

We then examined the effect of XN on alteration of redox status. Our results showed that the redox status markers (CD44v8-10 and phospho-p38MAPK) were reduced in the XN treatment group, and may lead to repression of CCA development via anti-oxidant activity. XN could protect cholangiocytes from oxidative stress during carcinogenesis; consequently, CD44v8-10 expression was not induced and phospho-p38MAPK was also decreased.

From our previous study, we found that the accumulation of CD44v8-10 led to the suppression of phospho-p38MAPK in transforming bile duct cells. The redox status regulation of CCA cells depends on the expression of CD44v8-10 to contribute the xCT function and is a link to the poor prognosis of patients (Thanee et al., 2016). Importantly, we found that a decrease of CD44v8-10 positive staining was seen during the transformation of bile duct epithelial cells from normal, hyperplasia, and dysplasia in XN treated hamster CCA model. In addition, a high phospho-p38MAPK positive signal staining was observed in both bile duct epithelial cells and surrounding inflammatory cells which was inversely correlated with the transformation of bile duct cells. These results suggested that XN could reduce expression of CD44v8-10 via the possible mechanism of oxidative damage inhibition (reduction of phospho-p38MAPK activation). Furthermore, it has been shown that XN could effectively suppress the growth of tumor and induced apoptosis in CCA cells and tumor inoculated mice (Dokduang et al., 2016). Currently, it has been reported that the oxidative stress could induce Wnt activation (Yoshida & Saya, 2014), and then triggered the transcription of CD44 (Ishimoto et al., 2010). Moreover, our group also showed an upregulation of Wnt/β-catenin signaling corresponded with the period of cholangiocarcinogenesis (Yothaisong et al., 2014). Based on this information, the possible mechanism is that the liver fluke infection-induced chronic inflammation causing oxidative stress leads to upregulation of Wnt/β-catenin signaling pathways and consequently increasing of CD44 expression, which may drive to CCA carcinogenesis. Hence, an inhibition of oxidative damage using XN could reduce expression of CD44v8-10 possibly via Wnt signaling pathway.

We next elucidated the effect of XN supplement on iron accumulation and TfR-1 expression. We found both decreasing iron accumulation and TfR-1 expression in bile ducts and inflammatory cells. Iron is an important determinant of ROS generation because an excess of iron can undergo redox reaction and stimulate free radical formation. H2O2-mediated toxicity was found to be iron mediated. TfR-1 knockdown has been shown to reduce iron uptake by 80% in the human hepatoma cell line (Herbison et al., 2009). Moreover, the aberrant expression of TfR-1 substantially contributed to the regulation of systemic iron levels and can play key roles in CCA development (Jamnongkan et al., 2017). TfR-1 has been reported to be a major molecule that responded to iron uptake mediated by insulin through hypoxia conditions (Biswas et al., 2013). Andriopoulos et al. (2007) found that exposure of cultured cells to sustained low levels of hydrogen peroxide that mimic its release by inflammatory cells lead to up-regulation of TfR-1. In steatotic livers, the saturation of β-oxidation by excess free fatty acids will ultimately lead to the generation of hydrogen peroxide, which in turn can be converted to highly reactive hydroxyl radicals in the presence of free iron via Fenton reaction (Videla et al., 2003). Indeed, there is strong evidence, from both in vitro and in vivo studies, that iron overload enhances oxidative stress (Brown et al., 2003; Cornejo et al., 2005; Kadiiska et al., 1995).

Sirius red staining was used in each hamster group to assess for levels of collagen which is related to fibrosis. It was revealed that a thick and extensive periductal fibrosis was found in the ON group which could be related to Ov-induced fibrosis caused by bile duct lesions that corresponded to the amount of collagen detected. The increased amount of collagen we detected around bile ducts is similar to findings by Prakobwong et al. (2009) who also showed that it is dependent on the duration of infection. In addition, we found that XN could reduce fibrosis from day 90 to 180 in another group of hamsters (XON group). Hence, XN has the potential to reduce periductal fibrosis in early stages of liver fluke infection as well as efficiently reducing fibrosis in long time treatment. We found that the reduction of fibrosis due to Ov infection was efficiently eliminated by PZ treatment in the ONP group. These results are similar to those of Pinlaor et al. (2009) who showed the effectiveness of PZ treatment for complete reduction of the fibrosis in liver tissues in order to prevent the progression of the parasite-induced diseases, and CCA development. Moreover, our study showed that the greatest reduction of fibrosis was in the XONP group compared to all other groups.

Praziquantel is well known as a potential drug for Ov treatment and removal of Ov. The cure rate of PZ treatment in opisthorchiasis can be as much as 100% effective in people (Bunnag & Harinasuta, 1981; Bunnag et al., 1984) and hamster (Duenngai et al., 2013). Decreasing liver fibrosis and egg granulomas has been revealed in the liver of patients with blood and liver fluke infections after PZ treatment (Pinlaor et al., 2010). In addition, it was demonstrated that promutagenic etheno-DNA and 8-oxodG adducts are most likely to increase the risk of Ov-infected patients to later develop CCA. A relationship between these adduct markers and disease causation is further supported by the protective effect of PZ against DNA damage in human (Dechakhamphu et al., 2010; Thanan et al., 2008). Similarly, this study suggests that CCA development might be suppressed after Ov treatment with PZ. Moreover, we have observed a high efficiency to suppress the progression of CCA in combination treatments between XN and PZ (XONP group). Other studies have indicated that the combination of curcumin (Charoensuk et al., 2011), and Thunbergia laurifolia (Wonkchalee et al., 2013) with PZ treatment improved the hepatobiliary system and could retard CCA progression.

Conclusions

In conclusion, we have demonstrated that PZ can reduce chronic inflammation in combination with XN treatment as well as suppressing oxidative stress possibly via the reduction of DNA damage, iron accumulation, periductal fibrosis and the alterations of intracellular redox lead to delayed CCA development. The proposed mechanism is shown in Fig. 8. Hence, XN in combination with PZ may prove to be useful chemopreventive agents for CCA prevention.

Figure 8 A possible chemopreventive effect of xanthohumol in combination with praziquantel.

Xanthohumol in combination with praziquantel could serve as potential anti-inflammation, anti-oxidant and antifibrotic agents in liver fluke-induced inflammation in relation to CCA development. Legend: ↓, regulate; ⊥, inhibit.

Supplemental Information

Data S1 Raw data

Click here for additional data file.

We would like to acknowledge Prof. Ross H. Andrews for editing the MS via Publication Clinic KKU, Thailand.

Additional Information and Declarations

Competing Interests

Author Contributions

Ethics

Data Availability

The authors declare there are no competing interests.

Wassana Jamnongkan conceived and designed the experiments, performed the experiments, analyzed the data, wrote the paper, prepared figures and/or tables, reviewed drafts of the paper.

Malinee Thanee conceived and designed the experiments, performed the experiments, analyzed the data, prepared figures and/or tables, reviewed drafts of the paper.

Puangrat Yongvanit and Raynoo Thanan conceived and designed the experiments, analyzed the data, reviewed drafts of the paper.

Watcharin Loilome and Tidarat Boonmars conceived and designed the experiments, contributed reagents/materials/analysis tools, reviewed drafts of the paper.

Phongsaran Kimawaha performed the experiments, prepared figures and/or tables, reviewed drafts of the paper.

Runglawan Silakit and Nisana Namwat contributed reagents/materials/analysis tools, reviewed drafts of the paper.

Anchalee Techasen conceived and designed the experiments, analyzed the data, wrote the paper, reviewed drafts of the paper.

The following information was supplied relating to ethical approvals (i.e., approving body and any reference numbers):

Animal study has been reviewed and approved by the Animal Ethics Committee of Khon Kaen University (AEKKU 23/2555), based on the ethics of animal experimentation of National Research Council of Thailand.

The following information was supplied regarding data availability:

The raw data has been provided as Data S1.

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
