# Peer review of "Antifibrotic effect of xanthohumol in combination with praziquantel is associated with altered redox status and reduced iron accumulation during liver fluke-associated cholangiocarcinogenesis"

_PeerJ, doi:10.7717/peerj.4281_

## Round 0.1 · original submission · Major Revisions

As you will see below, the two reviewers assigned think that the manuscript is poorly written and some important control experiments are lacking.

Reviewer 1 ·

Basic reporting

(i) Very unclear figures and its legends
Quality of figures seems to be good, but it is unclear what should be focused.
I recommend to use arrows and circles to point out important parts so that readers can focus.
It is also unclear how many sections the authors analyzed to make their conclusion. If they prepared immune-stained sections from all the treated mice (5 mice in each group?), that should be described.
(ii) No legends in supplementary figure.
What do weigh 1~8 mean in body weight?
Hamster No.5 seems to die 60 days after XON treatment. How did author measure liver weight per body weight of Hamster No. 5 at later stages?
(iii) It is very confusing because figure numbers are sometimes missing in Result section. Readers will not understand which figures they need to see in each description.

Experimental design

No treatment control (without Ov infection and NDMA) should be performed.
If authors have done, the results should be shown or clearly described.

Validity of the findings

(i) Quantitative and subsequent statistical analysis should be done to compare the difference between control and treated groups in immune-histochemical analysis. That will enable us to evaluate the effects of drugs more precisely.
Especially, it is difficult to judge whether there is any synergistic inhibitory effect of XN and PZ treatment on periductal fibrosis development. Evaluation of fibrosis stages (describe in methods) is not reflected on any figures.
(ii) XN and PZ treatment reduced several abnormalities, which are induced by Ov infection and NDMA treatment, such as DNA damage, CD44v8-10 expression, and fibrosis. Main concern is that only one of five hamsters in the ON group developed CCA. Although Ov-mediated ROS responses, inflammation and fibrosis result in CCA genesis, it seems very difficult to evaluate the effects of XN and PZ on CCA genesis under the condition used this study (just “suggest” will be better). In addition, it is not clear whether the provided figures in ON group are from CCA-developed hamster or from non-CCA hamster.
(iii) Figure 7 is confusing. For example, XN suppresses periductal fibrosis through reduction of the oxidative stress responses or independent of them? If the former is the case, what does the line from XN to periductal fibrosis mean?

Reviewer 2 ·

Basic reporting

no comment

Experimental design

no comment

Validity of the findings

no comment

Additional comments

In this manuscript, Jamnongkan et al. described the effects of xanthohumol (XN) in combination with praziquantel (PZ) on various aspects of liver fluke (Ov)-associated cholangiocarcinogenesis, including DNA damage, redox status change and periductal fibrosis induced by Ov and NDMA administration in hamsters. The overall experiments seem technically sound. However, the interpretation of the results is hard to follow (particularly the data on immunohistochemistry) due to the lack of quantitative data on immunohistochemistry as well as the lack of experiments with control hamsters.

A number of suggestions are as follows.
1) A major problem along the Results section is the lack of quantitative data on immunohistochemistry. The authors’ interpretation on the effects of XN treatment with or without PZ on histopathological findings, DNA damage, redox status, iron accumulation is largely based on their subjective impression, and no statistical analysis was performed to objectively evaluate the effectiveness of XN on liver fluke (Ov)-associated cholangiocarcinogenesis. They should classify the abnormal findings associated with (Ov)-associated cholangiocarcinogenesis, such as bile duct proliferation, hyperplasia, dysplasia of bile ducts, cholangio-fibrosis in each group during the progression of CCA (Figure 2). The graphs showing the number and/or levels of pathological abnormalities along time course should be included in the Figures.

2) In the same line with item 1, in order to validate their claim that either XN or PZ treatment could effectively reduce the accumulation of 8-oxodG, and that XNOP group could reduce DNA damage more effective than other groups (lines 253-255), the quantitative and the statistical analysis on the effect of XN with or without PZ on DNA damage should be performed. Also, the immunohistochemical data in Figure 4-6 requires quantitative measurements to validate their interpretations such as “no synergistic effect was observed in XNOP (line 265) or effective reduction in iron accumulation and TfR-1 expression for the XNOP as compared with other groups (lines 274-275).

3) In addition, the authors should include the data from untreated hamsters in Figures 1-6. Furthermore, the authors should indicate (by arrows or arrosheads) abnormalities such as the “white granules and small foci (line 229)” and “the obstruction of gall bladder” in the relevant Figures.

4) In the Introduction section, their description and the rationale on PZ treatment was insufficient.

5) I don’t’ understand the rational for measuring the CD44v8 to evaluate the redox status. Authors should describe the mechanisms as to how XN-induced alterations in oxidative stress would alter the expression of CD44v8-10.


6) In the Conclusion section, it is very vague whether their data indicate the effectiveness of sole XN treatment on Ov-associated CCA or their data indicate that the synergistic effect of XN in combination with PZ. These ambiguous conclusions could be derived from the lack of the quantification and rigorous statistical analysis.

7) There are numerous grammatical errors and typos.

8) Several papers are cited without being listed in the References, including Andriopoulos et al. (2007)

---

## Round 0.2 · Minor Revisions

The revised manuscript looks good. However, there remains minor concerns as indicated by Reviewer 1. Please address the concerns raised by this reviewer and send back the re-revised manuscript.

Reviewer 1 ·

Basic reporting

See general comments

Experimental design

See general comments

Validity of the findings

See general comments

Additional comments

1. In the first sentence of Results, "The liver per body weight of carcinogenesis in hamster groups either …." does not make sense. "The liver per body weight of hamster groups either …." will be better.

2. Body weight in supplementary figure is still confusing. If one column corresponds to the average of body weight from 5 mice, what does “Time 1-8” at each day (60, 90, 120, 180) mean?

3. Hamster No.5 seems to die 60 days after XON treatment. How did author measure liver weight per body weight of Hamster No. 5 at later stages?
Authors’ response: We measured liver per BW in the hamsters that were alive at each time point.

This is not the answer!
You put “death” into a column of hamster No.5, XON groups in the liver per BW sheet.
If the mouse died 60 days after the treatment, you can’t measure anything as in the case of ON groups.
However, you put values in 90, 120, 180 days about the hamster.

4. Showing figures are not necessary. However, the authors should, at least, describe the results from untreated groups anywhere in the manuscript.

Reviewer 2 ·

Basic reporting

In this manuscript, Jamnongkan et al. described the effects of xanthohumol (XN) in combination with praziquantel (PZ) on various aspects of liver fluke (Ov)-associated cholangiocarcinogenesis, including DNA damage, redox status change and periductal fibrosis induced by Ov and NDMA administration in hamsters. The overall experiments seem technically sound and their conclusions were well supported by various experiments with quantification analysis.

Experimental design

The manuscript has been significantly improved by the revision, with additional data that strengthen the authors' model.

Validity of the findings

All the concerns I raised about the previous version have been adequately.

Additional comments

The revised manuscript is much improved and I have no further comments.

---

## Round 0.3 · accepted · Accept

Your manuscript is now Accepted - congratulations.